# From Local Details to Global Context: Advancing Vision-Language Models with Attention-Based Selection

Lincan Cai [* 1]  Jingxuan Kang [* 2]  Shuang Li [3]  Wenxuan Ma [1]  Binhui Xie [1]  Zhida Qin [1]  Jian Liang [4]

## Abstract

Pretrained vision-language models (VLMs), e.g., CLIP, demonstrate impressive zero-shot capabilities on downstream tasks. Prior research highlights the crucial role of visual augmentation techniques, like random cropping, in alignment with fine-grained class descriptions generated by large language models (LLMs), significantly enhancing zero-shot performance by incorporating multiview information. However, the inherent randomness of these augmentations can inevitably introduce background artifacts and cause models to overly focus on local details, compromising global semantic understanding. To address these issues, we propose an **A**ttention-**B**ased **S**election (**ABS**) method from local details to global context, which applies attention-guided cropping in both raw images and feature space, supplement global semantic information through strategic feature selection. Additionally, we introduce a soft matching technique to effectively filter LLM descriptions for better alignment. **ABS** achieves state-of-the-art performance on out-of-distribution generalization and zero-shot classification tasks. Notably, **ABS** is training-free and even rivals fewshot and test-time adaptation methods. Our code is available at https://github.com/BIT-DA/ABS.

## 1. Introduction

Vision-language models (VLMs) (Radford et al., 2021; Alayrac et al., 2022; Jia et al., 2021; Xue et al., 2021) garner significant attention for their remarkable ability to perform zero-shot generalization across various downstream tasks. To better adapt VLMs, several prompt-tuning methods

*Equal contribution  [1]Beijing Institute of Technology [2]University of Illinois Urbana-Champaign [3]Beihang University [4]Kuaishou Technology. Correspondence to: Shuang Li <shuangliai@buaa.edu.cn>.

*Proceedings of the 42nd International Conference on Machine Learning*, Vancouver, Canada. PMLR 267, 2025. Copyright 2025 by the author(s).

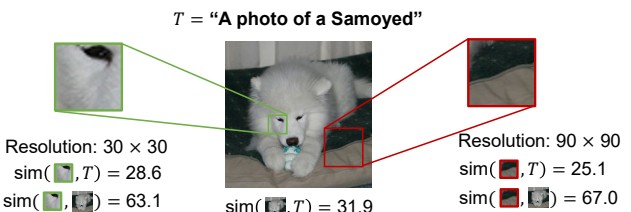

T = "A photo of a Samoyed"

Resolution: 30 × 30
sim(🟩, T) = 28.6
sim(🟩, 🟩) = 63.1

sim(🟩, T) = 31.9

Resolution: 90 × 90
sim(🟥, T) = 25.1
sim(🟥, 🟩) = 67.0

Figure 1: Random cropping for visual augmentation may capture background objects unrelated to the category (**red box**), which have lower similarity to the text compared to semantically meaningful objects (**green box**). Additionally, the randomness in crop size may result in background objects having a higher resolution than the main objects, leading to misjudgments when attempting to filter backgrounds based on image similarity.

(Zhou et al., 2022b;a; Khattak et al., 2023) introduce learnable text or image prompts while keeping VLM's pre-trained backbone fixed. Similarly, test-time adaptation (TTA) approaches (Shu et al., 2022; Feng et al., 2023; Karmanov et al., 2024) also achieve impressive results by finetuning VLMs online using test data. Although these methods prevent forgetting of pre-trained knowledge in VLMs by freezing the backbone and only finetuning learnable prompts or adapters, overfitting or a decline in generalization of VLMs still inevitably occurs (Ma et al., 2023; Zhu et al., 2023).

To maximize the generalization potential of VLM pretraining, recent studies show that manually designed prompts can significantly improve VLM performance on downstream tasks (Zhou et al., 2022b). However, the need for domain expertise and substantial time investment makes such approaches impractical for real-world applications. To address this issue, Pratt et al. (2023) uses category information as a prompt to guide large language models (LLMs) in generating fine-grained descriptions, effectively enriching the text prompt with detailed nuances. Furthermore, Li et al. (2024) find that text descriptions are often more precise for local image details. To improve alignment, they apply a random cropping operation to enhance image diversity, ensuring better matching between the image and text modalities.

Despite the benefits of the random cropping, this technique can inadvertently crop out background objects, leading to

misjudgments by the model. Although Li et al. (2024) assigns weights to cropped images based on their similarity to the original image, we find that the effectiveness of this similarity calculation is influenced by the crop size. As illustrated in Fig. 1, the green box provides a more semantically meaningful image while the red box cropped image is semantically meaningless. However, the red box, with its higher resolution, shows greater similarity to the original image. This suggests that using image similarity to assess the importance of a cropped image is not universally applicable. Therefore, before applying cropping, it is essential to focus on the primary objects within the image to avoid cropping backgrounds, which can mislead the model's judgment.

To tackle this, we propose an **A**ttention-**B**ased **S**election (**ABS**) method. The attention map of DINO (Caron et al., 2021) effectively highlights key objects within the image, making it ideal for guiding cropping. By leveraging this, we can target regions with higher attention values for cropping, ensuring that the focus remains on the main objects in the image. However, cropping at the image raw space alone can help the model focus on local object features but may result in the loss of global semantic information. As shown in Fig. 2, cropped images focusing on the bear's eyes may still be misclassified as a monkey due to the loss of global context, despite attending to local features. This situation highlights the limitation of image-level cropping in preserving the semantic integrity of the object's category.

To supplement the global semantic information of cropped images, we introduce feature selection that performing attention-guided cropping in feature space. By using the original images as input, we crop on the feature map before the model's final layer, extracting the crop features corresponding to the image-level crops. Since these features are derived from the original images, the model can retain global category information when extracting features. As shown in Fig. 2, the feature cropped from the feature map preserves the global semantic information of the bear, which would otherwise be lost in a purely image-level crop. By using these cropped features, we can enrich the global information of the corresponding cropped images, ultimately achieving better alignment between the image and the text descriptions, thereby enhancing VLM performance on downstream tasks. Additionally, we propose a soft matching method, which allows us to filter out text descriptions with low relevance to each crop, enabling more targeted matching.

In a nutshell, our contributions are summarized as follows: **(i)** We propose an **A**ttention-**B**ased **S**election (**ABS**) to guide the cropping process, focusing on the main objects in the image and minimizing the risk of cropping background objects. **(ii)** We introduce a feature selection that cropping at the feature map of the original image to supplement the cropped images with global information, ensuring that the

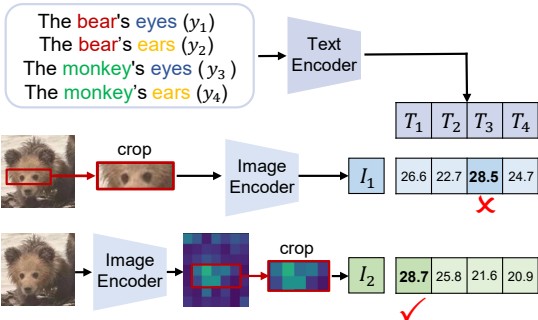

Figure 2: Similarity between the cropped image obtained in the image raw space and the cropped feature obtained at the feature map with the text descriptions. Although both crops can focus on the "eyes" as a local feature through cropping, the crop from the feature space retains the semantic information "bear", while the crop from the raw space misleads the model to identify it as "monkey".

model retains semantic understanding while focusing on local features. **(iii)** We propose a soft matching approach, enabling targeted matching of text descriptions to different patches. **ABS** achieves state-of-the-art performance in zero-shot classification and out-of-distribution datasets, even outperforming methods that require finetuning.

## 2. Related Work

### 2.1. Vision-Language Models

In recent years, Vision-Language Models (VLMs) make remarkable advancements in the domain of computer vision (Radford et al., 2021; Jia et al., 2021; Xue et al., 2021; Liu et al., 2023). These models acquire rich multimodal representations through joint pretraining on language and visual data, thereby outperforming traditional models (Dosovitskiy et al., 2020; He et al., 2016) that rely exclusively on image supervision. For instance, CLIP Radford et al. (2021) and ALIGN Jia et al. (2021) are trained on a dataset consisting of a large number of pairs of images and text. Further research, including BLIP (Li et al., 2022; 2023) and LLaVA (Liu et al., 2023), leverages CLIP and frozen LLMs as backbones, driving advancements in the VLM field.

Despite the impressive zero-shot capabilities and transferability demonstrated by VLMs, these models often overlook task-specific nuances, which can lead to suboptimal performance on downstream tasks (Zhou et al., 2022b). Consequently, effectively harnessing their representation capabilities presents a significant challenge. This study seeks to address this limitation by proposing a novel Attention-Based Selection mechanism from local details to global context.

### 2.2. Adapt VLMs To Downstream Tasks

The performance of pretrained Vision-Language Models (VLMs) in downstream tasks is significantly influenced by

the design of text prompts (Radford et al., 2021). Prompts can either be hand-crafted for specific tasks (Zhang et al., 2021; Gao et al., 2024) or learned automatically during the fine-tuning process (Zhou et al., 2022b;a). For better performance, the former approach often necessitates distinct prompt designs tailored to varying tasks and datasets. In contrast, the latter approach involves optimizing a continuous set of prompt vectors within the language branch of the model, enhancing alignment with the specific task. However, these techniques typically require few-shot data from the downstream task to effectively train the prompts. This process can be time-consuming and costly.

Menon & Vondrick (2022) as well as Pratt et al. (2023) illustrate the effectiveness of enhancing textual representations. This enhancement is achieved by integrating insights from large language models (LLMs) (Brown et al., 2020), which enable the automatic generation of descriptions tailored to specific classes. WCA (Li et al., 2024) suggests that local visual areas, obtained through random cropping, can be cross-aligned with more detailed descriptions by constructing a similarity matrix using a pre-trained Visual Language Model (VLM). However, the randomness inherent in these augmentations can introduce background artifacts and cause the model to overemphasize local information, potentially compromising its global semantic understanding. In contrast, our study employs DINO's (Caron et al., 2021) attention maps to guide data augmentation in both raw space and feature maps, effectively enhancing the model's ability to capture and integrate global semantic information.

### 2.3. Attention Guided Study

Local features of an image often contain finer details. Region-CLIP (Zhong et al., 2022) aims to focus the model on these local features through data augmentation. RedCircle (Shtedritski et al., 2023) also shows that encircling an object with a red circle can effectively direct a model's attention to that specific area. As we all know, attention, as a key component of transformer models, weights the relationships between image patches, identifying main objects. Thus, attention maps are effective tools for guiding focus on local features. FALIP (Zhuang et al., 2025) incorporates foveal attention within the image, allowing the model to transition more smoothly between the focal area and the background. In addition, ACEN (Chen et al., 2022a) generates attention maps to crop and randomly erasing regions to force the model to focus on key areas, ProxyCLIP (Lan et al., 2024) combines features from VFMs with CLIP through Proxy Attention, which enhances the prominence of primary objects, and zero-seg (Rewatbowornwong et al., 2023) proposes to balance global and local contexts within CLIP's attention layers by analyzing attention values to estimate region-wise saliency. However, these methods either fail to focus the model on local object features and lack the capability to

focus on localized features of individual objects or fail to address the subsequent loss of global context. In contrast, our approach leverages DINO's attention map, known for highlighting key objects, not only highlights local object characteristics but also preserves crucial semantic information, offering a more comprehensive solution, which can fully match fine-grained text descriptions, thereby enhancing CLIP's zero-shot performance.

## 3. Method

### 3.1. Preliminary

**Problem setting.** An image classification task involves an image space $\mathcal{X}$ and a label space $\mathcal{Y}$, where $\mathcal{Y}$ is a set of classname corresponding to each image, such as $\mathcal{Y} = \{\text{cat, dog}, \ldots, \text{car}\}$. The goal of a zero-shot classification task is to adapt pretrained VLMs to a downstream classification task without additional training. In a pretrained VLM, we demote $f$ as the image encoder and $g$ as the text encoder, which transforms input images $x$ and labels $y$ into a shared feature space of dimension $d$. Next, we will introduce the zero-shot classification capabilities of the CLIP model and discuss recent methods for generating visual and text prompts.

**Zero-shot classification of CLIP.** CLIP (Radford et al., 2021) is a VLM pretrained on 400 million image-text pairs using contrastive learning. It performs zero-shot classification by computing the cosine similarity between a given image and a set of labels. The scoring function is:

$$\text{sim}(x, y) = \cos(f(x), g(y)), \tag{1}$$

where $x$ is the given image, $y$ is one of the candidate labels, and $cos$ represents cosine similarity. A higher score indicates closer semantic alignment The predicted label $y^*$ is the one with the highest score for $x$ among all $y \in \mathcal{Y}$. The original CLIP constructs input text using hand-crafted prompts like $P_{\text{hc}} = $ "a photo of a $\{y\}$." Enhanced performance was achieved by manually designing 80 diverse prompts.

**Zero-shot classification using visual and text prompts.** Recent work (Pratt et al., 2023) utilizes category information as prompts to guide LLMs in generating detailed descriptions. For a classname $y \in \mathcal{Y}$, the descriptions generated are $y^{\text{llm}} = \{\text{LLM}(y)\}_{i=1}^{M}$, where $M$ is the number of descriptions. Building on this approach, (Li et al., 2024) propose a visual prompting method using random cropping aimed at generating fine-grained image regions that align better with $P_{\text{llm}}$. The score function for (Li et al., 2024) is defined as:

$$\text{sim}_{wca}(x, y) = \sum_{i=1}^{N} \sum_{j=1}^{M} w_i v_j \text{sim}(x_i, y_j^{\text{llm}}), \tag{2}$$

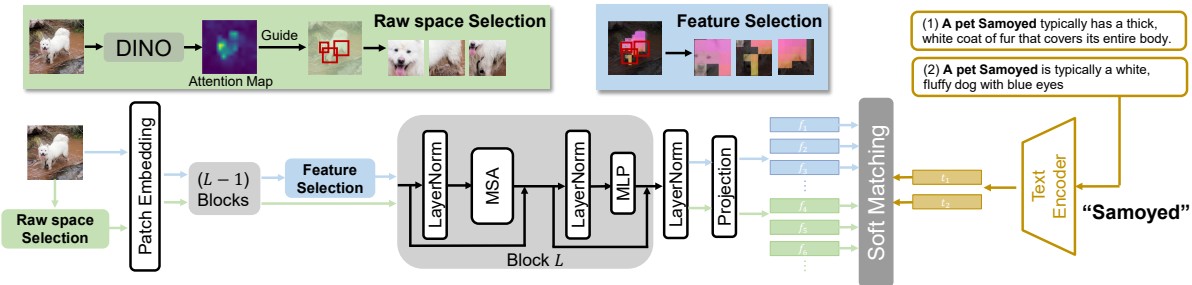

Figure 3: Framework overview. **Raw space selection:** We use DINO's attention map to guide image cropping, avoiding the inclusion of background objects. **Feature selection:** The original image is used as input and performs cropping on the feature map corresponding to the fine-grained selection before the final layer, to preserve global semantic information. **Soft matching:** We calculate a weight matrix to filter out irrelevant text descriptions for each crop, enabling better alignment.

where $w_i$ and $v_i$ are weights for filtering irrelevant images and descriptions, and $N$ indicate the number of crops. These approaches enhance image-text alignment, improving CLIP's zero-shot performance.

### 3.2. Attention-Based Raw Space Selection

Random cropping on images can yield more refined regions, but due to its randomness, the position and size of the cropped areas are uncertain, potentially cropping out background objects unrelated to the category. Although (Li et al., 2024) attempts to mitigate this with $w_i$, our analysis in Fig. 1 shows that $w_i$ is also influenced by crop size, making it difficult to effectively filter out background objects.

Therefore, we propose an attention-based method to guide image cropping named **A**ttention-**B**ased **S**election (**ABS**), aiming to avoid cropping background objects. Since DINO's (Caron et al., 2021) attention map is widely recognized for effectively capturing the main objects in an image, we use the attention map from the last transformer layer of DINO model for image $x$, denoted as $A \in \mathbb{R}^{P \times P \times h}$, where $P$ and $h$ represents the number of patches and attention heads respectively. We average attention maps from all heads:

$$\tilde{A} = \frac{1}{h} \sum_{i=1}^{h} A_i, \qquad (3)$$

where $A_i$ denotes the attention map of the $i$-th attention head. We then sort values in $\tilde{A}$ and select the top-$k$ patches:

$$p_{\text{top-}k} = \{p_{i_1}, p_{i_2}, \ldots, p_{i_k}\} \text{ where } p_{i_1} > p_{i_2} > \cdots > p_{i_k}, \qquad (4)$$

where $p_i$ represents the value corresponding to the $i$-th patch in the $\tilde{A}$. For the selected top-$k$ patches, we apply the softmax to obtain the probability of each patch being selected:

$$\text{Prob}(p_i) = \frac{\exp(p_i)}{\sum_{j=1}^{k} \exp(p_j)} \quad \text{for } i \in \{1, 2, \ldots, k\}. \qquad (5)$$

Based on the computed probability distribution, we sample the top-$k$ patches $N$ times. The sampling process can be represented by $\{p_{s_1}, p_{s_2}, \ldots, p_{s_N}\} \sim \text{Sampling}(p_{\text{top-}k}, \text{Prob}(p_i), N)$, where Sampling() refers to performing $N$ samples according to the probability distribution of the top-$k$ patches.

For each sampled patch, assuming its center position is $c_i$, we randomly select a crop size centered at $c_i$ and perform the cropping operation to obtain the final fine-grained selection:

$$p(x) = \{x_i = \phi(x, c_i, s) | i = 1, \ldots, N\}, \qquad (6)$$

where $\phi$ is the cropping operation and crop size $s = (\text{rand}(\alpha, \beta)W, \text{rand}(\alpha, \beta)H)$. Finally, we can obtain local features from raw space selection $F_{rs}(x) = f(x_i)_{i=1}^{N}$. By leveraging Dino's attention map, we ensure that the center of each crop is focused on the main object within the image, while the crop size is random. This approach guarantees both fine-grained character and diversity of the cropped images, preventing backgrounds from being cropped.

### 3.3. Attention-Based Feature Selection

By guiding image cropping with the attention map, we enable the model to focus more on the object's local features, resulting in better alignment with the fine-grained text description. However, as shown in Fig. 2, we find that while the crop helps the model focus on local features, it may also lead to the loss of global semantic information. Therefore, we propose an attention-based feature selection method, which supplements the lost global information by performing a crop operation on the original image's features, corresponding to the raw space selection.

We take the original image $x$ as input and extract features $F_{\text{mid}} = f_{l-1}(x)$ just before the final transformer layer, where $l$ is the number of transformer layers. Since the original image serves as the initial input, the extracted features at this stage retain global semantic information. We then perform the same cropping operation on the feature

map as the fine-grained attention-based selection, cropping out the corresponding $N$ feature maps:

$$p_{\text{fea}}(\tilde{x}) = \{\tilde{x}_i = \phi(F_{\text{mid}}, c_i, s) | i = 1, \ldots, N\}. \quad (7)$$

These cropped feature maps are resized back to the original size using *bicubic* interpolation and then reintroduced into the model to obtain the final features of feature selection $F_{fs}$. Through attention-based raw space and feature selection, for each image, we obtain $N$ features from raw space selection and corresponding $N$ features from feature selection, which preserve global information alongside local focus.

### 3.4. Soft Matching

At this point, we obtain the final feature $F$ which contains the fine-grained features $F_{rs}$ and holistic features $F_{fs}$ corresponding to an image. But intuitively, not all descriptions are suitable for each cropped image. For example, if the cropped image is of a dog's eye but the text description refers to its ears or tail, the image and description do not match. Forcing such a mismatch into the final score could interfere with the model's output. To address this, we propose a soft matching approach. First, we compute the similarity $\text{sim}_d^i$ between each crop and all descriptions across categories:

$$\text{sim}_d^i = \cos(F_i, g(y_j^c)) \quad \text{for } j = 1, \ldots, M; \ c = 1, \ldots, K \quad (8)$$

where $K$ is the number of categories. The description weight vector $w_d^i$ is obtained via softmax:

$$w_d^i = Softmax(\text{sim}_d). \quad (9)$$

The final score function is:

$$\text{sim}_{\text{abs}}(x, y) = \sum_{i=1}^{2N} \sum_{j=1}^{M} w_d^i \cos(F_i, g(y_j^{\text{llm}})). \quad (10)$$

The algorithm of ABS is illustrated in Alg. 1. For additional details of our method, please refer to the Appendix A.1.

## 4. Experiment

In this section, we first introduce datasets and baselines that relevant to our work, and our implementation details. Then, we validate the effectiveness of **ABS** on two benchmark with three different backbones, comprising a total of 10 datasets. Finally, through a series of analytical experiments including component ablation, parameter sensitivity and visualization and so on, we showcase the superiority of each module within **ABS** when compared to alternative approaches.

**Datasets.** In alignment with recent studies (Li et al., 2024), we conduct evaluations across two established benchmarks: **(1)** out-of-distribution generalization and **(2)** zero-shot classification. For the out-of-distribution generalization, we evaluate our methods on the variants of ImageNet.

---

**Algorithm 1** Attention-Based Selection

---

**Require:** input image $\boldsymbol{x} \in \mathbb{R}^{H \times W \times 3}$, DINO sampled patches $\mathcal{P} = \{p_i\}_{i=1}^{N}$, Crop size bounds $\alpha, \beta \in (0, 1)$
1: mid_fea = CLIP($\boldsymbol{x}$, layer = $l - 1$)
2: **for** each patch $p \in \mathcal{P}$ **do**
3:      Sample crop size: $c_{\text{size}} \sim \mathcal{U}(\alpha, \beta)$
4:      **# Raw Space Selection:**
5:      $\boldsymbol{x}_{\text{crop}} = \phi(\boldsymbol{x}, p.\text{center}, c_{\text{size}})$
6:      $\boldsymbol{f}_{\text{raw}} = \text{CLIP}(\boldsymbol{x}_{\text{crop}})$
7:      raw_crops.append($\boldsymbol{f}_{\text{raw}}$)
8:      **# Feature Space Selection:**
9:      $\boldsymbol{f}_{\text{crop}} = \phi(\text{mid\_fea}, p.\text{center}, c_{\text{size}})$
10:     $\boldsymbol{f}_{\text{resize}} = \text{Interpolate}(\boldsymbol{f}_{\text{crop}})$
11:     $\boldsymbol{f}_{\text{fea}} = \text{CLIP.final\_layer}(\boldsymbol{f}_{\text{resize}})$
12:     fea_crops.append($\boldsymbol{f}_{\text{fea}}$)
13: **end for**
14: com_fea = Concat(raw_crops $\oplus$ fea_crops)

---

ImageNetV2 (Recht et al., 2019) presents a distribution shift that simulates real-world scenarios, while ImageNet-Sketch (Wang et al., 2019) consists of black-and-white sketches that challenge models to recognize objects based on outlines rather than photographic details. ImageNet-A (Hendrycks et al., 2021b) includes naturally occurring images that serve as adversarial examples, testing the robustness of classification models against atypical inputs. Lastly, ImageNet-R (Hendrycks et al., 2021a) features a diverse set of images that vary in style, blurriness, geographic location, and camera operation, aiming to evaluate the adaptability of models to different visual conditions. For the zero-shot classification benchmark, we adhere to the methodology outlined in (Menon & Vondrick, 2022). This benchmark encompasses several datasets, including ImageNet (Deng et al., 2009), a comprehensive object recognition dataset; CUB (Welinder et al.), which focuses on fine-grained bird classification; Oxford Pets (Parkhi et al., 2012), an animal classification dataset; DTD (Cimpoi et al., 2014), a texture recognition dataset; Food101 (Bossard et al., 2014), which contains a diverse range of food images; and Place365 (Zhou et al., 2017), designed for scene classification tasks.

**Baselines.** In the context of the zero-shot classification and out-of-distribution (OOD) generalization benchmark, we evaluate our method using three different backbones against several baseline approaches. The baselines are as follows: (i) CLIP (Radford et al., 2021): Utilizes a photo of class as the text prompt for classification; (ii) CLIP-E (Radford et al., 2021): An enhanced variant of CLIP that employs an ensemble of hand-crafted prompts; (iii) CLIP-D (Menon & Vondrick, 2022): Leverages LLMs for generating descriptive text associated with the classes; (iv) CuPL (Pratt et al., 2023): Improves upon CLIP-D by

Table 1: The Top-1 accuracy (%) of the out-of-distribution generalization benchmark using three different CLIP backbones (ViT-B/32, B/16, and L/14), the bold values highlight the highest accuracy in the table. $\sigma$ represents the standard deviation and $\triangle$ indicates the improvement of our method over the top-performing baseline, which is underlined.

| Method | ImageNet | | | ImageNet-V2 | | | ImageNet-R | | | ImageNet-S | | | ImageNet-A | | | Average | | |
|---|---|---|---|---|---|---|---|---|---|---|---|---|---|---|---|---|---|---|
| | B/32 | B/16 | L/14 | B/32 | B/16 | L/14 | B/32 | B/16 | L/14 | B/32 | B/16 | L/14 | B/32 | B/16 | L/14 | B/32 | B/16 | L/14 |
| CLIP | 62.05 | 66.74 | 73.48 | 54.79 | 60.83 | 67.88 | 66.24 | 73.98 | 85.40 | 40.78 | 46.10 | 57.81 | 29.56 | 47.75 | 68.83 | 50.68 | 59.08 | 70.68 |
| CLIP-E | 63.37 | 68.37 | 75.52 | 55.97 | 61.90 | 69.85 | 69.33 | 77.68 | 87.82 | 42.29 | 48.25 | 59.60 | 31.61 | 49.93 | 70.77 | 52.51 | 61.23 | 72.71 |
| CLIP-D | 63.01 | 68.04 | 75.03 | 56.25 | 61.32 | 68.88 | 66.29 | 74.69 | 86.08 | 40.83 | 46.91 | 57.99 | 30.57 | 48.79 | 69.16 | 51.39 | 59.95 | 71.43 |
| Waffle | 63.30 | 68.12 | 75.31 | 55.72 | 61.71 | 69.35 | 67.34 | 75.85 | 86.96 | 41.48 | 48.30 | 58.76 | 31.23 | 50.31 | 70.28 | 51.81 | 60.86 | 72.13 |
| CuPL | 64.37 | 69.61 | 76.62 | 57.09 | 63.27 | 70.72 | 68.36 | 77.06 | 87.69 | 42.46 | 49.00 | 59.00 | 31.15 | 50.69 | 71.85 | 52.69 | 61.93 | 73.18 |
| WCA | 66.84 | 71.08 | 77.32 | 59.81 | 64.71 | 71.46 | 69.47 | 78.06 | 88.22 | 43.86 | 50.18 | 59.77 | 35.58 | 56.13 | 75.63 | 55.11 | 64.03 | 74.48 |
| ABS | 67.74 | 71.92 | 77.62 | 61.41 | 66.19 | 72.07 | 72.16 | 79.57 | 88.77 | 44.37 | 50.54 | 60.16 | 41.79 | 61.80 | 77.80 | 57.49 | 66.00 | 75.28 |
| $\sigma$ | 0.03 | 0.03 | 0.02 | 0.04 | 0.09 | 0.05 | 0.06 | 0.03 | 0.03 | 0.01 | 0.02 | 0.05 | 0.10 | 0.05 | 0.03 | - | - | - |
| $\triangle$ | +0.90 | +0.84 | +0.30 | +1.60 | +1.48 | +0.61 | +2.69 | +1.51 | +0.55 | +0.51 | +0.36 | +0.39 | +6.21 | +5.67 | +2.17 | +2.38 | +1.97 | +0.80 |

Table 2: The Top-1 accuracy (%) of the zero-shot classification benchmark using three different CLIP backbones (ViT-B/32, B/16 and L/14). The bold values highlight the highest accuracy in the table and underlining indicates the second-best results.

| Method | ImageNet | | | CUB | | | Oxford Pets | | | DTD | | | Food101 | | | Place365 | | |
|---|---|---|---|---|---|---|---|---|---|---|---|---|---|---|---|---|---|---|
| | B/32 | B/16 | L/14 | B/32 | B/16 | L/14 | B/32 | B/16 | L/14 | B/32 | B/16 | L/14 | B/32 | B/16 | L/14 | B/32 | B/16 | L/14 |
| CLIP | 62.05 | 66.74 | 73.48 | 51.21 | 56.01 | 62.12 | 85.04 | 88.14 | 93.24 | 42.93 | 42.98 | 52.61 | 82.60 | 88.40 | 92.55 | 38.51 | 39.27 | 39.63 |
| CLIP-E | 63.37 | 68.37 | 75.52 | 52.74 | 56.16 | 62.53 | 87.38 | 89.1 | 93.62 | 43.83 | 45.27 | 55.43 | 83.93 | 88.83 | 93.07 | 39.28 | 40.30 | 40.55 |
| CLIP-D | 63.01 | 68.04 | 75.03 | 52.69 | 57.08 | 63.26 | 84.46 | 87.52 | 93.30 | 44.20 | 46.17 | 55.05 | 84.12 | 88.85 | 93.03 | 39.9 | 40.34 | 40.55 |
| Waffle | 63.30 | 68.12 | 75.31 | 52.04 | 56.89 | 62.27 | 85.50 | 86.51 | 91.55 | 44.68 | 44.63 | 54.31 | 83.98 | 89.06 | 93.33 | 39.47 | 40.76 | 40.89 |
| CuPL | 64.37 | 69.61 | 76.62 | 49.76 | 56.42 | 62.15 | 87.03 | 91.14 | 94.33 | 47.50 | 50.53 | 60.59 | 84.20 | 88.98 | 93.37 | 39.08 | 39.83 | 40.77 |
| WCA | 66.84 | 71.08 | 77.32 | 56.91 | 59.78 | 65.24 | 89.89 | 92.23 | 94.66 | 49.39 | 52.79 | 61.78 | 86.40 | 90.01 | 93.96 | 40.66 | 41.43 | 42.23 |
| ABS | 67.74 | 71.92 | 77.62 | 57.42 | 61.01 | 67.06 | 90.39 | 92.64 | 94.88 | 51.65 | 54.26 | 61.80 | 85.66 | 89.69 | 93.05 | 41.22 | 41.88 | 42.27 |

generating higher-quality descriptions; (vi) Waffle (Roth et al., 2023): Substitutes LLM-generated descriptions with randomly generated character and word descriptions; (v) WCA (Li et al., 2024): Implements random cropping and visual-text cross-alignment to enhance classification performance. In addition, we conduct a comparative analysis of our method against several fine-tuning approaches within the context of OOD generalization benchmarks. Specifically, we evaluate our method alongside CoOp (Zhou et al., 2022b), CoCoOp (Zhou et al., 2022a), UPT (Zang et al., 2022), ProGrad (Zhu et al., 2023), KgCoOp (Yao et al., 2023), TPT (Shu et al., 2022), DiffTPT (Feng et al., 2023), TDA (Karmanov et al., 2024) and GDA (Wang et al., 2024).

**Implementation details.** Our experiments are conducted using the CLIP model with various backbones, including ViT-B/32, ViT-B/16, and ViT-L/14. All experiments are performed on an NVIDIA 4090 GPU. Our method incorporates four key parameters: the crop lower and upper bound $(\alpha, \beta)$, the top importance of the patch $(K)$, and the number of crops $(N)$. In our study, we maintain consistent parameters across all architectures and datasets. Specifically, we set $\alpha = 0.5$, $\beta = 0.9$, $K = 20$, $N = 60$, and $M = 50$.

### 4.1. Overall Results

**Out-of-distribution generalization.** In the out-of-distribution generalization benchmark, we compare our method with six zero-shot baselines using three different

CLIP backbones: ViT-B/16, ViT-B/32, and ViT-L/14. As shown in Table 1, **ABS** achieved state-of-the-art results across all datasets and backbones. On individual datasets, we improved top-performing baselines by up to 6.21%, and on average, we achieved a 2.38% improvement, demonstrating the effectiveness of our method.

**Zero-shot classification.** In the zero-shot classification experiment, we used three different CLIP backbones: ViT-B/16, ViT-B/32, and ViT-L/14, and compared **ABS** with six zero-shot baselines. As shown in Table 2, **ABS** achieved the best results on five out of six datasets, demonstrating its superiority on fine-grained category datasets.

**Comparing with finetuning methods.** The results in Table 3 compare **ABS** with a series of fine-tuning approaches on the Out-of-Distribution generalization benchmark, using the ViT-B/16 backbone. As shown in Table 3, **ABS** achieved the best results on all datasets except ImageNet, where it slightly lagged behind UPT. Notably, **ABS** is a training-free zero-shot approach, yet it outperforms fine-tuning methods like few-shot learning and test-time adaptation, highlighting the effectiveness and superiority of our approach.

### 4.2. Analytic Experiments

**Ablation study.** Our component ablation study is presented in Table 4, where we use ViT-B/16 as the backbone and perform experiments on three datasets: Ima-

Table 3: The Top-1 accuracy (%) of the out-of-distribution generalization benchmark using ViT-B/16 as the CLIP backbone compared with some finetuning methods, such as fewshot and TTA methods. "Tuned" means the model is finetuned on ImageNet and tested on target datasets.

| Method | Tuned? | Source | Target | | | | Average |
| --- | --- | --- | --- | --- | --- | --- | --- |
| | | ImageNet | ImageNet-V2 | ImageNet-R | ImageNet-S | ImageNet-A | |
| CoOp (Zhou et al., 2022b) | ✓ | 71.51 | 64.20 | 75.21 | 47.99 | 49.71 | 61.72 |
| CoCoOp (Zhou et al., 2022a) | ✓ | 71.02 | 64.07 | 76.18 | 48.75 | 50.63 | 62.13 |
| UPT (Zang et al., 2022) | ✓ | **72.63** | 64.35 | 76.24 | 48.66 | 50.66 | 62.51 |
| ProGrad (Zhu et al., 2023) | ✓ | 72.24 | 64.73 | 74.58 | 47.99 | 49.39 | 61.79 |
| KgCoOp (Yao et al., 2023) | ✓ | 71.20 | 64.10 | 76.70 | 48.97 | 50.69 | 62.33 |
| TPT (Shu et al., 2022) | ✓ | 69.70 | 64.30 | 73.90 | 46.40 | 53.67 | 61.59 |
| DiffTPT (Feng et al., 2023) | ✓ | 70.30 | 65.10 | 75.00 | 46.80 | 55.68 | 62.58 |
| TDA (Karmanov et al., 2024) | ✓ | 69.51 | 64.67 | **80.24** | **50.54** | 60.11 | 65.01 |
| CuPL (Pratt et al., 2023) | ✗ | 69.61 | 63.27 | 77.10 | 48.80 | 50.77 | 61.91 |
| GDA (Wang et al., 2024) | ✗ | 72.23 | 65.04 | 76.97 | 48.96 | 50.51 | 60.37 |
| WCA (Li et al., 2024) | ✗ | 71.08 | 64.71 | 78.06 | 50.18 | 56.13 | 64.03 |
| **ABS** | ✗ | 71.92 | **66.19** | 79.57 | **50.54** | **61.80** | **66.00** |

geNet(Deng et al., 2009), DTD(Cimpoi et al., 2014), and ImageNet-V2(Recht et al., 2019), reporting top-1 accuracy. We take CuPL(Pratt et al., 2023) as the baseline and progressively add components of our method for the ablation study. Specifically, $F_{rs}$ refers to the use of attention-based raw space selection, $F_{fs}$ refers to the use of attention-based feature selection, and Soft-M refers to the application of soft matching for alignment. From Table 4, we observe that both $F_{rs}$ and $F_{fs}$ individually improve the results by approximately 0.3% compared to the CuPL. The small improvement observed when using the raw space and feature selection individually because: Using either selection alone may lead to excessive focus on either local or global information. When both selection methods are employed simultaneously, they complement each other, leading to an improvement of approximately 1.3%. In addition, the use of cropping will focus on specific local regions. While this enables better alignment with LLM descriptions that match the currently focused regions, it weakens the alignment for those unrelated to these regions. Consequently, without soft matching to filter irrelevant descriptions, many unrelated descriptions would adversely affect the current crop's alignment. When combined with soft matching, the results improve by about 1.5%. In the final row, **ABS**, which integrates both local and global information along with soft matching, achieves a 2.98% improvement, highlighting the importance of the complementary nature of local details and global context and the significance of filtering irrelevant descriptions.

**Different visual augmentation ways.** Both prior work (Li et al., 2024) (Jia et al., 2022) and our experiments demonstrate the importance of visual augmentation in enhancing the model's zero-shot generalization capability. Therefore, we conduct experiments using various visual augmentation methods in both raw space and feature space. In the

Table 4: Ablation Study on three datasets: ImageNet, DTD, and Imagenet-V2 using ViT-B/16 as the backbone. The bold values highlight the highest accuracy in the table. The first row represents the (Pratt et al., 2023) which only uses LLM descriptions for alignment, and the last row represents our method. △ indicates the average improvement of these three datasets compared to the (Pratt et al., 2023).

| Component | | | Datasets | | | △ (Avg.) |
| --- | --- | --- | --- | --- | --- | --- |
| $F_{rs}$ | $F_{fs}$ | Soft-M | ImageNet | DTD | Imagenet-V2 | |
| - | - | - | 69.61 | 50.53 | 63.27 | - |
| ✓ | - | - | 69.34 | 51.52 | 63.56 | +0.33 |
| - | ✓ | - | 69.34 | 51.81 | 63.10 | +0.28 |
| - | - | ✓ | 70.03 | 52.89 | 63.68 | +1.06 |
| ✓ | ✓ | - | 70.32 | 52.66 | 64.34 | +1.30 |
| ✓ | - | ✓ | 70.98 | 52.72 | 65.51 | +1.93 |
| - | ✓ | ✓ | 70.31 | 53.88 | 63.92 | +1.56 |
| ✓ | ✓ | ✓ | **71.92** | **54.26** | **66.19** | **+2.98** |

raw space, augmentations included mask, highlight, redcircle (Shtedritski et al., 2023), and crop. Here, the mask sets non-selected area pixels to zero, and the highlight increases the brightness of the selected area. In the feature map, augmentations included mask, highlight (fea.), highlight (attn.), and crop. Highlight (fea) and highlight (attn) refer to highlighting in the feature map and attention map respectively. As shown in Table 5, crop outperform other augmentation methods, whereas mask performs poorly in both raw and feature space. We believe this is because CLIP's pretraining augmentation primarily uses crops, making it less adaptable to other augmentation methods. We aim to explore more VLMs and related visual augmentations in the future.

**Parameter sensitivity.** In this subsection, we analyze the sensitivity of three hyperparameters on the ImageNet dataset

Table 5: Ablation study on different visual augmentation ways in raw space and feature space. highlight (fea.) and highlight (attn.) represent the highlight in the feature map and attention map respectively.

| raw space | ImageNet | feature space | ImageNet |
|---|---|---|---|
| mask | 53.65 | mask | 62.72 |
| highlight | 66.44 | highlight (fea.) | 65.03 |
| redcircle | 62.63 | highlight (attn.) | 67.37 |
| crop | **70.98** | crop | **70.31** |

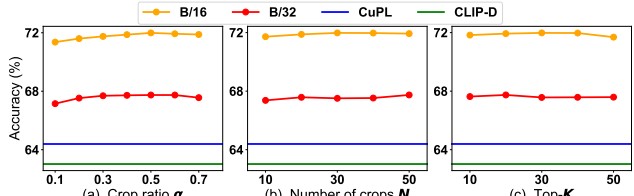

Figure 4: The sensitivity of three hyperparameter: crop ratio $\alpha$, number of crops $N$ and value of Top-$k$ on on ImageNet dataset using different CLIP backbones, ViT-B/16 and ViT-B/32, comparing with two baselines CLIP-D and CuPL.

using two different CLIP backbones, ViT-B/16 and ViT-B/32. The first hyperparameter is the crop ratio. We fix $\beta$ and test the sensitivity of $\alpha$, ranging from $[0.1, 0.7]$. As shown in Fig. 4(a), the model's performance is not sensitive to the crop ratio, as it generally forms a horizontal line. We observe that, unlike WCA(Li et al., 2024), where performance increases with crop ratio, the model performs better with a smaller crop ratio in some cases. For example, the result with a crop ratio of $0.5$ outperforms the result with $0.7$. We attribute this to the inclusion of $F_{\mathrm{fs}}$, which supplements global semantic information to the cropped image. As a result, when the crop ratio is small, the cropped image focuses more on local features without losing semantic context, leading to better performance.

The second parameter is $N$, which is the number of crops. As shown in Fig. 4(b), our results remain stable across all values of $N$, even when $N$ is as small as 10. This demonstrates the effectiveness of our attention-based selection method, which ensures that even with a small number of crops, the performance is not compromised by randomly cropping background objects. Notably, when $N$ is small, the efficiency of our method improves, meaning that we can reduce the inference time without sacrificing accuracy.

The third parameter is top-$k$, we select the top-$k$ patches based on the attention values from DINO's attention map, using their attention values as probabilities for sampling. The sampled patches are then used as centers for cropping. As shown in Fig. 4(c), our results are minimally affected by the choice of $k$, with the optimal performance achieved at $k = 20$. This is because we sample based on the probabilities of the patches, so even with a larger $k$, the smaller attention values lead to lower sampling probabilities, which minimizes the impact on the results. This demonstrates the robustness of our approach.

**Visualization.** Due to the randomness of random cropping, it is unavoidable to crop background objects, which can mislead the model's classification. Therefore, we use DINO's (Caron et al., 2021) attention map to guide the image cropping. As shown in Fig. 5, DINO's attention map effectively locates key objects in the image. We center the

crop around the top-$k$ values of the attention map, ensuring the focus remains on the main objects in the image. The cropped images in Fig. 5 contain features from different regions of the main objects while avoiding background objects that are completely unrelated to the image category. Moreover, we find that DINO's attention map is superior to CLIP's. While both DINO and CLIP focus on the main objects in an image, DINO's focus is more distributed, avoiding the extreme values seen in CLIP. This results in cropped images with greater diversity, capturing more aspects of objects. Additional crop visualization experiments can be found in the appendix.

**Effect of attention map guiding.** In this paper, we experimentally show that guiding cropping with DINO's attention map improves the quality of cropped images and enhances model performance. In Table 6, we investigate the impact of different attention maps by comparing random cropping with three other attention map-guided cropping methods. We use CLIP ViT-B/16 as the backbone and experiment with random crop, CLIP attention map, DINO-S/16, and DINO-B/16 guidance across three datasets. As shown in Table 6, random crop yields the worst results due to its inherent randomness, which often crops background objects, misleading the model's judgment. Among the three attention map-guided methods, DINO-B/16 performs the best. This is because, based on visualization analysis in Fig. 5, DINO's attention map outperforms CLIP's. Therefore, we can conclude that using stronger attention maps better guides cropping, ultimately improving model's zero-shot performance.

Table 6: Comparison of different attention map guiding methods, including random cropping, attention map of CLIP-B/16, DINO-S/1,6, and DINO-B/16.

| | ImageNet | CUB | DTD | ImageNet-a | Average |
|---|---|---|---|---|---|
| Random Crop | 69.60 | 53.58 | 52.40 | 54.31 | 57.47 |
| CLIP-B/16 | 70.37 | 59.82 | 54.05 | 60.11 | 60.84 |
| DINO-S/16 | 71.77 | 60.92 | 54.12 | 59.45 | 61.66 |
| DINO-B/16 | **71.92** | **61.01** | **54.26** | **61.80** | **62.25** |

Table 7: Ablation study on using feature space selection in different transformer layers.

| Layer_id | 1 | 2 | 3 | 4 | 5 | 6 | 7 | 8 | 9 | 10 | 11 |
|---|---|---|---|---|---|---|---|---|---|---|---|
| acc. (%) | 66.14 | 66.42 | 66.26 | 66.41 | 66.82 | 66.99 | 67.10 | 67.23 | 67.44 | 67.54 | **67.74** |

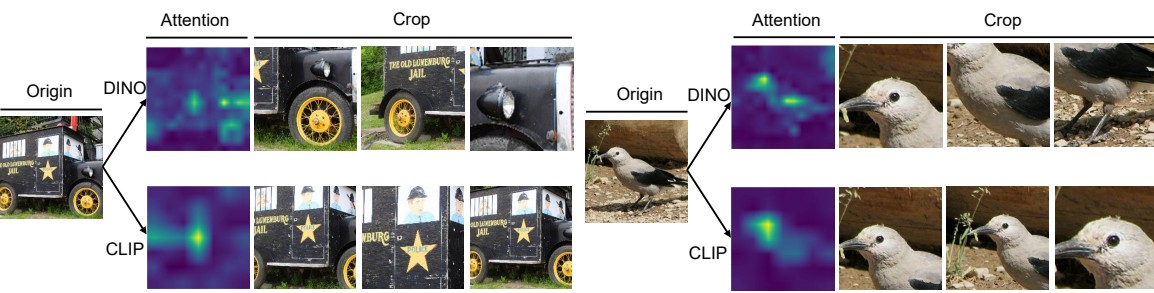

Figure 5: The visualization of the DINO and CLIP attention map and the cropped images guided by the attention maps.

**Integrating with other VLMs.** To further validate the effectiveness and transferability of our method, we conducted additional experiments on ImageNet using multiple VLM backbones, including ALIGN (Jia et al., 2021), AltCLIP (Chen et al., 2022b), and GroupViT (Xu et al., 2022). While these models share general similarities with CLIP (Radford et al., 2021), they exhibit distinct architectural designs or pretraining configurations. As shown in Table 8 (where "CLIP" denotes directly using the single-image and "CLIP prompt" features obtained from different VLMs for classification), the consistent performance gains across all benchmarks demonstrate the superiority and robustness of our approach. Additional results of more advanced VLP model like BLIP-2 (Li et al., 2023) can be found in Appendix A.3.

Table 8: Comparison with different methods across various VLMs on ImageNet.

| VLM | CLIP | CLIP-E | CLIP-D | Waffle | CuPL | WCA | ABS |
|---|---|---|---|---|---|---|---|
| ALIGN | 65.24 | 65.79 | 65.08 | 65.22 | 66.24 | 66.77 | **67.85** |
| AltCLIP | 73.79 | 74.86 | 74.48 | 74.29 | 75.74 | 76.20 | **76.85** |
| GroupViT | 37.11 | 42.72 | 40.10 | 42.42 | 44.53 | 45.27 | **46.96** |

**Feature selection in different layers.** Table 7 presents the results of performing feature space selection at different layers of the transformer in CLIP. It can be observed that the model's accuracy generally increases with deeper layers. We attribute this to the model's improved extraction of global semantic features at deeper layers. Cropping features at shallow layers yields similar to not using $F_{\text{fs}}$, indicating that the model has not captured category information, making it almost identical to cropping in the raw space. This suggests that the resulting feature $F_{\text{fs}}$ lacks global contextual information, as shallow-layer cropping tends to overemphasize local patterns. In contrast, cropping at deeper layers captures global representations, complementing the

raw space selection to form more comprehensive features, thereby enhancing model performance.

## 5. Conclusion

In this paper, we propose **A**ttention-**B**ased **S**election (**ABS**) from local details to global context, which leverages DINO's attention map to guide cropping in both raw space and intermediate feature maps. This approach yields crops that focus on local object characteristics as well as those containing global semantic information. Additionally, we filter text descriptions for each crop using soft matching to achieve better feature matching. Our method achieves state-of-the-art results across two benchmarks including ten datasets using different backbones.

**Limitation and future work.** During the course of this research, we identify some limitations in the current version and directions for future improvement: Firstly, our method improves model performance by leveraging stronger attention maps. However, the attention map of a single model can be limited. We look forward to exploring segmentation models, such as SAM, to further guide the cropping process. Secondly, we have found that crop-based augmentation is currently the most effective for CLIP. We aim to explore more diverse augmentation techniques to complement each other and provide richer features for CLIP. Thirdly, our method is currently limited to image-text modalities, and we plan to explore its applicability across additional modalities and different tasks.

## Impact Statement

This paper presents work whose goal is to advance the field of Machine Learning. There are many potential societal consequences of our work, none which we feel must be specifically highlighted here.

## Acknowledgements

This paper was supported by the National Natural Science Foundation of China (No. 62376026), Beijing Nova Program (No. 20230484296) and Beijing Nova Programme Interdisciplinary Cooperation Project.

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

# A. Appendix

## A.1. Algorithm of ABS

The algorithm of ABS is illustrated in Alg. 1. Firstly, we select $N$ patches of the raw space image from DINO's attention map using top-$k$ sampling. Next, we determine the crop size by randomly sampling between $\alpha$ and $\beta$ based on the positional centers of the selected $N$ patches. Using these centers and the derived crop sizes, we perform cropping operations in both the raw space (for images) and the feature space (for features), thereby obtaining crops that incorporate both local and global characteristics.

The specific operations and dimensional transformations in the feature space are as follows: feature selection is performed before the forward of the final transformer layer. Given input features with dimensions [bs, 197, 768], we first separate the [CLS] token ([bs, 1, 768]) from the remaining tokens ([bs, 196, 768]). The remaining tokens are reshaped into $2D$ size [bs, 14, 14, 768]. Based on DINO's attention map, we then select $N$ crops from this feature map. Each crops is interpolated to the original feature map size, and concatenated with the [CLS] token, reconstructing the feature into [bs, N, 197, 768]. This modified feature is fed into the final transformer layer. During this layer's forward, the [CLS] token interacts with the crops to capture diverse local features enriched with global semantic information.

## A.2. Visualization

As shown in Fig. 7, we select images from various datasets for visualization, including DINO's attention and the cropped images obtained from raw space selection guided by the attention map. It is evident that with effective guidance from the attention map, our cropped areas focus on the main objects in the images and capture different features of the objects.

## A.3. Additional Experiments

**Time cost of raw space selection and feature selection.**   As shown in Table 9, we calculate the time consumption for the "crop+preprocess" and "Encoding" stages. "crop+preprocess" includes the time for raw space selection, while "Encoding" covers feature selection. Although **ABS** takes more time than CLIP, the performance improvement is significant.

Table 9: The "Crop+Preprocess" and "Encoding" time cost of CLIP and **ABS** for different numbers of crops in seconds.

| Process Step | CLIP | N | | | | |
| --- | --- | --- | --- | --- | --- | --- |
| | | 10 | 20 | 30 | 40 | 50 |
| Crop+Preprocess | 0.0004 | 0.0011 | 0.0028 | 0.0036 | 0.0045 | 0.0060 |
| Encoding | 0.0004 | 0.0094 | 0.0168 | 0.0244 | 0.0322 | 0.0340 |
| Total | 0.0008 | 0.0105 | 0.0196 | 0.0280 | 0.0367 | 0.0400 |
| Accuracy | 66.74 | 71.72 | 71.92 | 71.98 | 71.97 | 71.93 |

**More advanced VLP model.**   We employ the Blip2ForImageTextRetrieval (Li et al., 2023) model architecture and compare ABS with other baseline methods (where "CLIP" denotes directly using the single-image and "CLIP prompt" features obtained from BLIP-2 for classification). As shown in the Table 10, ABS outperforms all other approaches across two different datasets, demonstrating its effectiveness and adaptability.

Table 10: Comparison with other methods using BLIP-2 as backbone on DTD and ImageNet-A datasets.

| Base model: BLIP-2 | CLIP | CLIP-E | CLIP-D | Waffle | CuPL | WCA | ABS |
| --- | --- | --- | --- | --- | --- | --- | --- |
| DTD | 42.55 | 46.54 | 51.70 | 45.64 | 50.74 | 54.47 | **56.12** |
| ImageNet-A | 59.96 | 58.08 | 62.99 | 61.43 | 63.51 | 72.79 | **74.39** |

## A.4. Failure Case Discussion

The relatively modest performance of our method on the Food101 dataset can be attributed to the following factors: The Food101 differs from conventional multi-object datasets, it contains inherently multi-label images but provides only single-

label annotations. For instance, an image labeled as "french fries" may actually contain multiple objects (e.g., fries, steak, and salad), where non-target objects could occupy a larger visual proportion than the labeled subject (as shown in Fig. 6). Because all these objects fall within the predefined label set, the single-label assignment introduces ambiguity. These inherent properties could cause our method to identify unlabeled object categories within the images.

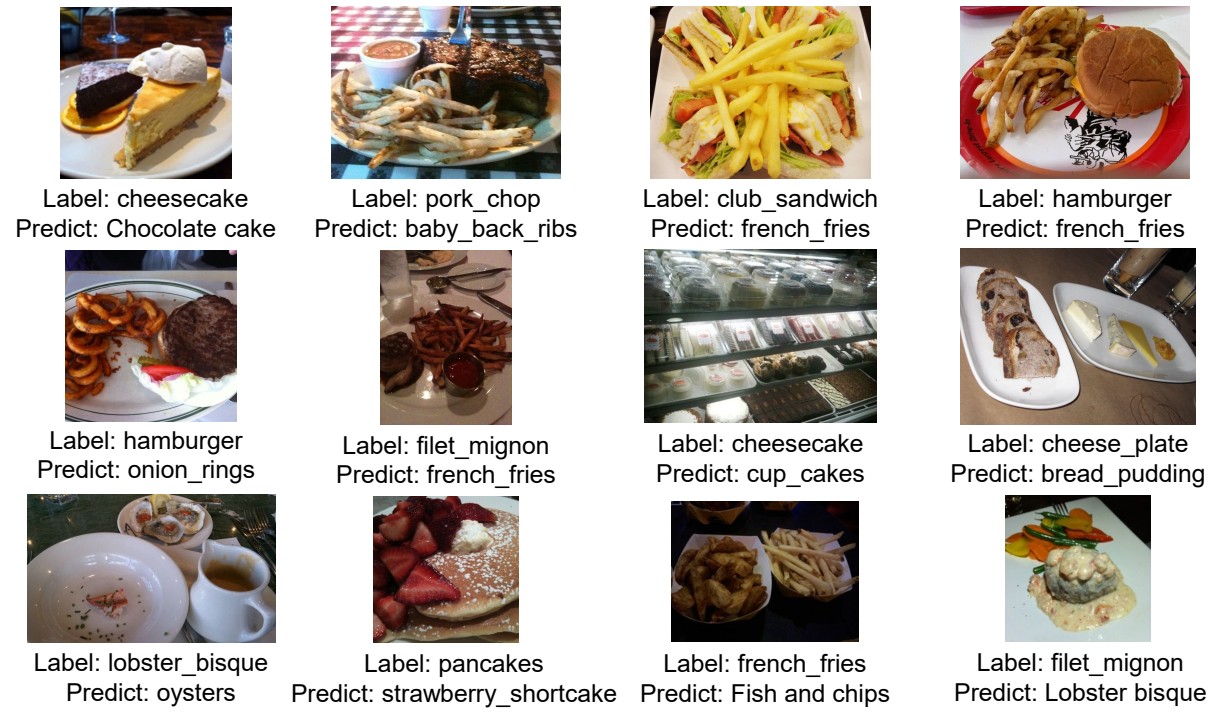

Figure 6: Examples of Food101 dataset.

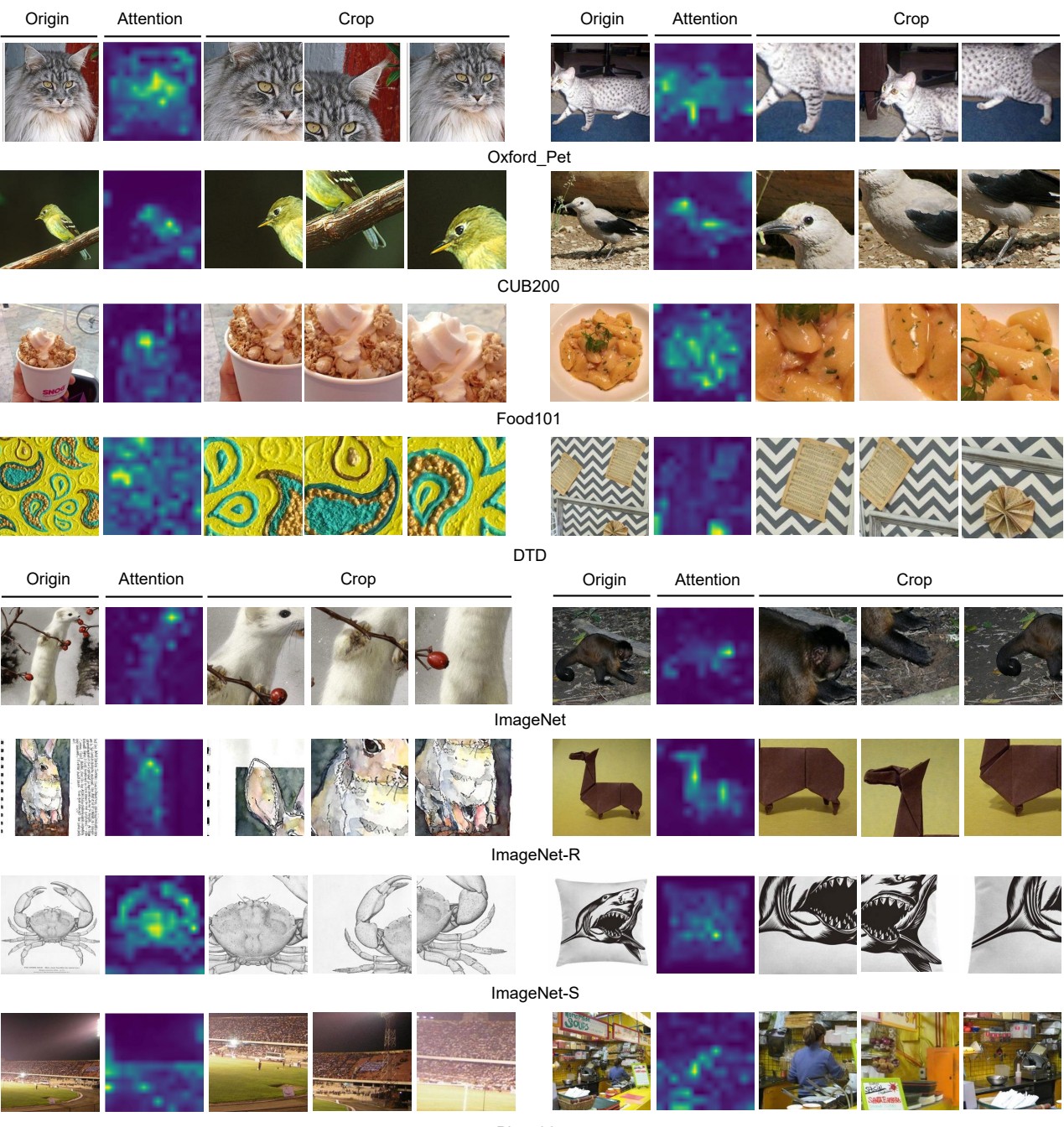

Figure 7: The visualization of the DINO attention map and the cropped images guided by attention maps.

