# OpenReview forum: "From Local Details to Global Context: Advancing Vision-Language Models with Attention-Based Selection"
_ICML.cc/2025/Conference — ICML 2025 poster_

### Official Review · Reviewer_iUpr · 2025-03-02

**Overall Recommendation:** 3

**Summary:**

The paper introduces **Attention-Based Selection (ABS)**, a method to enhance vision-language models (VLMs) like CLIP by addressing limitations of random cropping, which often introduces background noise and compromises global semantic understanding. ABS leverages **DINO's attention maps** to guide cropping in **Raw Image Space** and **Feature Space**.  Additionally, a **soft matching** technique filters irrelevant text descriptions for each crop, improving alignment between visual and textual modalities.

The main results indicate that ABS achieves **state-of-the-art performance** on 10 datasets across two benchmarks (zero-shot classification and out-of-distribution generalization), outperforming methods like WCA and CuPL.  Besides, it matches or surpasses fine-tuning approaches (e.g., CoOp, TPT) without requiring additional training data in the out-of-distribution generalization benchmark. Ablation studies also validate the necessity of each component, with ABS improving accuracy by up to **2.98%** over baselines.

The main findings are that introducing DINO feature maps to select key regions for cropping is effective, and cropping both the original images and the intermediate feature maps is crucial.

**Claims And Evidence:**

Yes,  the claims made in the submission are supported by clear and convincing evidence.

**Essential References Not Discussed:**

No.

**Experimental Designs Or Analyses:**

Yes, I have reviewed all the experiments mentioned in the Section 4. I have identified the following issues:
1. The experimental details in Table 3 are not described clearly enough. For example, it is not clear how many shots are specifically used, or whether the whole data from ImageNet are used for fine-tuning. Besides, fine-tuning the model with the data from ImageNet itself may actually reduce the generalization ability of the model on other distributions of ImageNet, because the fine-tuning process may causes the model to overfit the distribution of the original ImageNet. Therefore, the baselines that this method needs to compare with should be models fine-tuned using the few-shot datasets from target datasets like ImageNet-V2 and ImageNet-R.

**Methods And Evaluation Criteria:**

Yes.

**Other Comments Or Suggestions:**

NA.

**Other Strengths And Weaknesses:**

**Strengths:**
1. The method proposed in this paper is simple, effective, and intuitive.
2. The paper is clearly written and quite readable.

**Weaknesses:**

I have two main concerns regarding the ABS method proposed in this paper:
1. The ABS method strongly depends on the quality and accuracy of the Attention Map generated by DINO. Therefore, when the attention map of DINO is of poor quality (for example, it focuses on the background or misidentifies objects), this method may have a negative impact on the object classification accuracy of the model.
2. There is a significant increase in inference time overhead. For single - image classification, this method changes the number of images for feature extraction from 1 to 2N, thus significantly reducing the corresponding inference speed. This situation is also shown in Table 8. Although Table 8 indicates that the performance is greatly improved compared to the original CLIP, I think ABS should not only compare it with the original CLIP but also with methods like CuPL. Since these methods also only need to infer one image, the performance improvement compared to them may not be significant enough to ignore the growing reasoning overhead.

**Questions For Authors:**

1. What is the dimension of $F_{mid}$ in Equation 7? The specific process of Feature Map interpolation could be described in more detail.
2. How can we maximize the avoidance of the negative impact of inaccurate attention maps of DINO on the method?
3. Soft Matching essentially weights the image-text similarities of different images and texts. The goal of this weighting is to increase the relative ratio of the image-text similarities between similar and dissimilar image-text pairs. Have the authors tried other weighting methods, such as the approach similar to cross-modal late interaction in FILIP [1]? It would be better if there were some exploratory experiments in this regard.

[1] Yao, Lewei, et al. "FILIP: Fine-grained Interactive Language-Image Pre-Training." International Conference on Learning Representations.

**Relation To Broader Scientific Literature:**

The method proposed in this paper can help VLMs improve their performance in tasks such as image recognition.

**Theoretical Claims:**

No, because this paper doesn't has proofs for theoretical claims.

---

> ### Author Rebuttal · Authors · 2025-03-31
>
> **Q1**: Experimental details in Table 3.
>
> **A1:** Thanks. The results of baselines in Table 3 follow WCA protocol: Tuning-based methods are 16-shot source-trained and target-evaluated for OOD generalization. Notably, our method requires no fine-tuning, operating in a zero-shot manner on both source and target datasets and even surpasses fine-tuned methods.
>
> Regarding your concern, we include in the table below the results of Coop fine-tuned directly on target data. Since our method functions as a plug-and-play module without requiring fine-tuning, integrating it with Coop further enhances performance.
>
> ||ImageNet-S|ImageNetv2
> |-|-|-
> |Coop ft. on tar.|49.79|70.59
> |Coop ft. on tar. + ABS|**51.47**|**72.63**
>
> **Q2:** About negative impact of DINO.
>
> **A2:** Thanks.
> 1. DINO establishes a new contrastive learning framework that learns superior visual representations. Its exceptional attention maps, which have gained prominence for reliably pinpointing primary objects across diverse datasets, provide the foundation for our feature selection mechanism. And also a lot of works [2][3] using DINO for assistance. This reliability in object localization motivates our strategic adoption of DINO's attention maps to guide our selection process.
> 2. We observe that DINO's attention maps consistently localize semantically relevant object regions in multiple datasets. While occasional inaccuracies may occur, our adaptive crop_ratio enhances diversity in the cropped regions to mitigate such cases. Furthermore, our soft matching strategy effectively downweights irrelevant image crops, thereby reducing potential negative impacts from DINO's misalignments.
> 3. We observe that different attention heads in DINO may focus on distinct regions. In our paper, we simply average all attention heads. To further mitigate potential negative effects from DINO, we use attention maps from heads with higher variance, ensuring the selection maintain strong focus on salient objects. The following table shows the results demonstrate nearly identical performance to our original method. This suggests that ABS already achieves low error rates in attention map selection, making further error reduction less noticeable.
>
> Regarding more explicit filtering of negative attention maps, we will conduct further research.
>
> ||ImageNet|ImageNet-A
> |-|-|-
> |Ours w. diverse head|71.96|61.86
> |Ours w. mean head|71.92|61.80
>
> ref:
>
> [2] Zero-guidance segmentation using zero segment labels. CVPR 2023.
>
> [3] Proxyclip: Proxy attention improves clip for open-vocabulary segmentation. ECCV 2024.
>
> **Q3:** About inference time overhead.
>
> **A3:** Thanks. When processing and encoding images, we do need to augment a single image into 2N crops. However, our hyperparameter sensitivity experiments reveal that the results are not sensitive to the num_crop. Therefore, we can reduce the value of N to lower time costs, while achieving average improvements of approximately **2%** on Zero-shot classification and **5%** on OOD datasets compared to CuPL, both of which represent significant gains. Additionally, we can follow the approach of WCA by pre-storing image features prior to inference. This ensures that the final similarity computation stage incurs comparable time costs to CuPL, while delivering substantially enhanced performance.
>
> ||encoding time|zero-shot avg. acc.|OOD avg. acc.
> |-|-|-|-
> |CuPL|0.0008|39.83|61.93
> |ABS (N=10)|0.0105|41.65|65.59
>
> **Q4:** About dimension in Equation 7.
>
> **A4:** Thanks. The specific process of interpolation please refer to **our answer A2 to Reviewer dz4u** and the pseudo-code is as follows:
> ```
> # s_p: sampled patches from DINO attention
> raw_crops = []
> fea_crops = []
> mid_fea = CLIP(x, layer=l-1)
> for p in s_p:
>     c_size = random(α, β)
>     c_img = crop_img(x, p.center, c_size)
>     raw_fea = CLIP(c_img)
>     raw_crops.append(raw_fea)
>
>     c_fea = crop_fea(mid_fea, p.center, c_size)
>     r_fea = interpolate(c_fea, original_size)
>     f_fea = CLIP.final_layer(r_fea)
>     fea_crops.append(f_fea)
> com_fea = concatenate(raw_crops + fea_crops)
> ```
>
> **Q5:** About negative impact of DINO.
>
> **A5:** Thanks. Please refer to **A2**.
>
> **Q6:** Comparison with other weighted methods.
>
> **A6:** Thanks. FILIP [1] proposes a token-level matching approach for VLM pretraining. However, since its pretrained model is not publicly available, we cannot directly apply its token-level matching to VLMs that were not pretrained with this method. As an alternative, we experiment with the weight-based matching approach proposed in WCA and common entropy weighted methods and compared them with our soft matching. The results demonstrate the superior performance of our soft matching, indicating its enhanced capability to select more semantically aligned image-text pairs for effective matching.
>
> ||ImageNet|DTD|ImageNetV2
> |-|-|-|-
> |Ours w/ WCA weighted|71.22|53.36|65.23
> |Ours w/ entropy weighted|70.76|53.12|64.94
> |Ours|**71.92**|**54.26**|**66.19**

---

### Official Review · Reviewer_dz4u · 2025-03-11

**Overall Recommendation:** 2

**Summary:**

Recent studies have explored the use of multiple image crops obtained through random cropping, utilizing text descriptions generated by LLMs to assess the similarity between image and text embeddings for zero-shot classification tasks. This paper builds on that concept by addressing the noise caused by random cropping operations. It does this by selectively sampling crops and feature subsets that focus on the salient regions identified by the attention map from DINO. Experimental evaluations across several benchmarks show performance improvement.

**Claims And Evidence:**

The claim regarding uncertainty from random cropping in Figure 1 is not sufficiently valid. The only sample sizes provided are 30×30 and 90×90, which fall outside the interval (0.4, 0.9) used in this paper. At higher resolutions, the uncertainty introduced by the cropping operation may significantly decrease. Therefore, a more comprehensive empirical evaluation is necessary to clarify this claim.

**Essential References Not Discussed:**

- The clarity regarding this paper's position in recent attention-guided studies is still ambiguous (Sec.2.3).

**Experimental Designs Or Analyses:**

- The experiment results only demonstrate compatibility with CLIP.  Since the feature selection strategy ( $F_{fs}$ ) may not be model-agnostic, I'm curious whether the proposed pipeline can integrate with other VLMs. A related study, specifically WCA, has shown results using ALIGN, AltCLIP, and GroupViT (please refer to Table 10 of [1]). Conducting a similar experiment could provide valuable insights.

- The results shown in Table 4 of the ablation study indicate that the factors ( $F_{rs}$ ) and ($ F_{fs} $) contribute to performance degradation on some benchmarks. It appears that the main improvement in performance is primarily attributable to the soft-matching strategy, while the impacts of ($ F_{rs} $) and ($ F_{fs} $) remain unclear.

[1] Visual-Text Cross Alignment: Refining the Similarity Score in Vision-Language Models. ICML 2024.

**Methods And Evaluation Criteria:**

- The operation described in lines 208-211, which involves "reintroducing the interpolated cropped feature maps into the model," raises some concerns. Do the authors concatenate the original [CLS] token with the interpolated feature map as the input for the final attention residual block? If so, how can the [CLS] token effectively capture the local semantics associated with the cropped region?  A more detailed clarification or analysis would be appreciated.

**Other Comments Or Suggestions:**

n/a

**Other Strengths And Weaknesses:**

### Pros:

- The paper is well organized.

- The motivation is clear, and the method is simple and intuitive.

**Questions For Authors:**

n/a

**Relation To Broader Scientific Literature:**

- The proposed framework is mainly based on WCA [1], with its own contributions of handling the cropping uncertainty.

- Utilizing network feedback mechanisms such as attention to guide the cropping process [2] and zero-shot inference [3,4] is a well-explored technique. The attention-based strategy presented in this paper resembles those earlier studies.

[1] Visual-Text Cross Alignment: Refining the Similarity Score in Vision-Language Models. ICML 2024.

[2] Crafting better contrastive views for siamese representation learning. CVPR 2022.

[3] Proxyclip: Proxy attention improves clip for open-vocabulary segmentation. ECCV 2024.

[4] Zero-guidance segmentation using zero segment labels. CVPR 2023.

**Theoretical Claims:**

There is no theoretical analysis provided.

---

> ### Author Rebuttal · Authors · 2025-03-31
>
> **Q1:** Claims And Evidence.
>
> **A1:** Thanks. The purpose of employing cropping is to enable the model to focus on local features of objects, thereby achieving better alignment with certain LLM descriptions. However, random crop exhibits two inherent limitations: When using smaller crop sizes, it introduces randomness and compromises global semantic coherence; while with larger crop sizes, although uncertainty diminishes (as you mentioned), this comes at the cost of sacrificing the ability to concentrate on local features, instead providing more global information similar to using full images.
>
> Our method addresses this by enabling the model to eliminate randomness while preserving global semantics when focusing on local and pivotal features. Through ablations in the table below, we observe that our approach does not exhibit monotonically increasing accuracy with larger crop sizes. Conversely, it achieves superior results with smaller crop sizes, whereas WCA demonstrates an incremental pattern with increasing crop sizes. This contrast underscores the significance of our method's dual capability in eliminating randomness and maintaining global semantic integrity during local feature focusing, ultimately enhancing the model's optimal performance.
>
> |crop_ratio|0.3-0.9|0.4-0.9|0.5-0.9|0.6-0.9|0.7-0.9|0.8-0.9
> |-|-|-|-|-|-|-
> |WCA|70.80|70.89|70.95|70.99|71.03|**71.03**
> |ABS|71.75|71.87|71.92|**71.99**|71.88|71.84
>
> **Q2:** Clarification of feature crop operation.
>
> **A2:** Thanks. In our method, feature selection is performed before the forward of the final transformer layer. Given input features with dimensions [bs, 197, 768], we first separate the [CLS] token ([bs, 1, 768]) from the remaining tokens ([bs, 196, 768]). The remaining tokens are reshaped into 2D size [bs, 14, 14, 768]. Based on DINO’s attention map, we then select N crops from this feature map. Each crops is interpolated to the original feature map size, and concatenated with the [CLS] token, reconstructing the feature into [bs, N, 197, 768]. This modified feature is fed into the final transformer layer. During this layer's forward, the [CLS] token interacts with the crops to capture diverse local features enriched with global semantic information. We will supplement this detail in subsequent versions of our paper.
>
> **Q3:** Integrating with other VLMs.
>
> **A3:** Thanks. As shown in the table below, we apply our method to a broader range of VLMs (e.g. ALIGN, AltCLIP and GroupViT) in ImageNet and achieved superior performance compared to other approaches.
>
> |VLM|Waffle|CuPL|WCA|ABS
> |-|-|-|-|-
> |ALIGN|65.22|66.24|66.77|**67.85**
> |AltCLIP|74.29|75.74|76.20|**76.85**
> |GroupViT|42.42|44.53|45.27|**46.96**
>
> **Q4:** The impact of different components of our method.
>
> **A4:** Thanks. Please refer to **our answer A1 to Reviewer C2Mc.**
>
> **Q5:** Contribution of the work.
>
> **A5:** Thanks. Please refer to **our answer A1 to Reviewer bJ6A.**
>
> **Q6:** About other attention-based strategy methods.
>
> **A6:** Thanks. [2] employs heatmaps to localize target regions. However, it relies on dynamically updating bounding boxes during training, which is incompatible with our zero-shot scenario. Furthermore, while [2] can successfully locate target regions, it fails to address the preservation of global information after cropping, which may lead to inter-class confusion in the model.
>
> [3] combines features from VFMs with CLIP through Proxy Attention, which enhances the prominence of primary objects. [4] proposes to balance global and local contexts within CLIP's attention layers by analyzing attention values to estimate region-wise saliency. However, these methods fail to focus the model on local object features and lack the capability to focus on localized features of individual objects. In contrast, our approach not only highlights local object characteristics but also preserves crucial semantic information, offering a more comprehensive solution.
>
> We implement the core modules of [3] and [4] within our framework and conduct experiment on the ImageNet dataset. As shown in the table below, our method demonstrates superior performance, validating the advantage of our approach in simultaneously focusing on local features while preserving global semantic information. A comprehensive comparison with methods [2]-[4] will be presented in the subsequent version of our paper.
>
> ||ImageNet
> |-|-
> |Proxyclip [3]|70.36
> |zero-seg [4]|68.37
> |Ours|**71.92**
>
> **Q7:** The clarity regarding this paper's position.
>
> **A7:** Thanks. The key insight of ABS compared to previous attention-based methods lies in two aspects: not only do we leverage attention-guided cropping in the raw space to focus on local regions while eliminating randomness, but more importantly (and to the best of our knowledge) we are the first to implement attention-guided feature space selection to complement global semantics. This dual-space cropping enables the crops to better align with LLM descriptions.

---

### Official Review · Reviewer_bJ6A · 2025-03-12

**Overall Recommendation:** 3

**Summary:**

The paper proposes an Attention-Based Selection (ABS) method to improve zero-shot classification and out-of-distribution generalization capabilities of vision-language models (VLMs). ABS leverages DINO’s attention maps to guide the cropping of images, thus preventing random crops from focusing on irrelevant background areas. The method also introduces feature-level cropping to supplement global semantic context. Finally, a soft matching mechanism filters large language model (LLM)-generated text descriptions to improve visual-textual alignment. The authors report state-of-the-art results on multiple benchmarks, demonstrating the effectiveness of ABS.

**Claims And Evidence:**

The claims made by the authors are supported by the experimental results, and the motivation behind the proposed approach is sound.

**Essential References Not Discussed:**

The paper provides a comprehensive summary of recent relevant literature.

**Experimental Designs Or Analyses:**

The comparisons across multiple widely recognized datasets and three different CLIP backbones (ViT-B/32, ViT-B/16, ViT-L/14) clearly support the viewpoint.

**Methods And Evaluation Criteria:**

The proposed method, including the use of DINO's attention maps for guiding cropping, feature-level cropping, and soft matching of textual descriptions, is conceptually clear, well-motivated. The evaluation on standard benchmarks aligned with current practices in the vision-language modeling literature.

**Other Comments Or Suggestions:**

1. In Equation (1) and (10), it is recommended to use an upright font for "cos" to maintain consistency throughout the paper.

2. There are inconsistencies in the capitalization of certain headings.

**Other Strengths And Weaknesses:**

**Strengths**
1. The paper is clearly written and provides strong motivation for its proposed method.

2. Extensive experiments validate the effectiveness of the proposed method, and fair comparisons were conducted.

3. The visualizations clearly reflect the authors' intentions.

**Weaknesses**
1. Although the method is effective, its contribution to the field is relatively minor, merely proposing a small modification to an existing problem.

2.  Table 6 (Lines 425-539) is not properly cited; instead, numerical results are presented directly in the text.

**Questions For Authors:**

1. How were the hyperparameters set, and why were they unified across all datasets?

2. How sensitive is the method to the precision of soft matching? Would the performance drop significantly if the matching precision is imperfect or noisy (e.g., descriptions slightly mismatching the cropped areas)?

**Relation To Broader Scientific Literature:**

The paper is effectively situated within the broader literature of vision-language modeling, specifically highlighting recent advances such as WCA, CuPL, and methods employing attention-based augmentation.

**Theoretical Claims:**

The paper does not explicitly present theoretical proofs or formal theoretical claims.

---

> ### Author Rebuttal · Authors · 2025-03-31
>
> **Q1:** About contribution.
>
> **A1:** Thanks. Our core contribution resides not in proposing incremental adjustments to established frameworks, but rather in advancing systematic methodologies that demystify stochastic factors while enhancing holistic semantic comprehension for this research domain.
>
> Firstly, the adoption of attention-based mechanisms to mitigate cropping randomness plays a pivotal role in zero-shot classification task, enabling consistent and robust performance improvements.
>
> Moreover, our key insight lies not merely in employing attention-based selection to focus on critical regions, but also in being the first (to the best of our knowledge) to perform **feature cropping** that complement raw-space crops by integrating global semantic information. This dual mechanism proves crucial for aligning with LLM descriptions, while also establishing a novel paradigm for future research in multimodal alignment.
>
> Additionally, our approach demonstrates high flexibility, enabling seamless integration with various advanced VLMs (please refer to **our answer A2 to Reviewer C2Mc and A3 to Reviewer dz4u**) to enhance their performance through plug-and-play adaptation.
>
> **Q2:** Table 6 is not properly cited.
>
> **A2:** Thanks for pointing it out. We will fix it in the future version of our paper.
>
> **Q3:** Maintain “cos” consistency.
>
> **A3:** Thanks for pointing it out. We will fix it in the future version of our paper.
>
> **Q4:** Capitalization consistency.
>
> **A4:** Thanks for pointing it out. We will fix it in the future version of our paper.
>
> **Q5:** About hyperparameters set.
>
> **A5:** Thanks. Our method involves three critical hyperparameters: crop_ratio, num_crops, and top-k, which we consistently set to 0.5, 60, and 20 respectively across all datasets. This unified configuration serves dual purposes: First, it demonstrates our approach's capability to achieve consistent performance improvements without dataset-specific tuning, and second, it facilitates convenient application in industrial deployments. As evidenced by the hyperparameter sensitivity analysis presented in the table below or in the Figure5 in our paper, our method exhibits remarkable robustness to parameter variations. In fact, alternative configurations (e.g. crop ratio=0.6, num_crops=30 and top-k=30) might potentially yield superior results to those reported in our paper. Nevertheless, we maintain fixed parameters throughout all experiments to ensure fair comparative evaluation and rigorous empirical validation of the proposed methodology.
>
> | crop_ratio | 0.3 | 0.4 | 0.5 | 0.6 | 0.7 | 0.8 |
> | --- | --- | --- | --- | --- | --- | --- |
> | ABS | 71.75 | 71.87 | 71.92 (reported in paper) | **71.99** | 71.88 | 71.84 |
> | **num_crops** | **10** | **20** | **30** | **40** | **50** | **60** |
> | ABS | 71.72 | 71.88 | **71.98** | 71.97 | 71.93 | 71.92 (reported in paper) |
> | **top-k** | **10** | **20** | **30** | **40** | **50** | **60** |
> | ABS | 71.83 | 71.92 (reported in paper) | **71.98** | 71.97 | 71.69 | 71.56 |
>
> **Q6:** About soft matching.
>
> **A6:** Thanks. The soft matching mechanism is proposed to addresses the inherent mismatch issue in CLIP's original matching strategy, specifically the "description-crop misalignment" phenomenon where textual descriptions partially deviate from cropped image regions. This problem arises because our selection process emphasizes local object features - compared to using full images, these focused crops may induce partial mismatches with LLM descriptions during semantic alignment. The soft matching resolves this through adaptive weighting: it suppresses irrelevant LLM descriptions while amplifying semantically aligned ones. As demonstrated in the table below, the ablation studies confirm that removing soft matching causes performance degradation, verifying its critical role in mitigating mismatch effects. Notably, applying soft matching to WCA further improves accuracy, empirically proving its general effectiveness in filtering noisy descriptions.
>
> |  | ImageNet | DTD | ImageNetV2 |
> | --- | --- | --- | --- |
> | Ours w/o soft matching  | 71.22 | 53.36 | 65.23 |
> | Ours | **71.92** | **54.26** | **66.19** |
> | WCA | 71.08 | 52.79 | 64.71 |
> | WCA w/ soft matching | **71.37** | **53.60** | **65.67** |

---

### Official Review · Reviewer_C2Mc · 2025-03-12

**Overall Recommendation:** 3

**Summary:**

This paper introduces ABS, a training-free Attention-Based Selection method that uses Vision-Language Pretraining (VLP) model’s attention maps (e.g., DINO and CLIP) to guide cropping in both raw image and feature space, effectively integrating local details with global semantic context via soft matching to achieve SoTA performance in some zero-shot classification tasks.

**Claims And Evidence:**

The primary claims may not be supported by convincing evidence. Specifically, the effectiveness of the attention‐based raw space and feature selection components is not fully supported by the ablation study provided. In Table 4, these components contribute only a marginal average improvement (around +0.3%). It makes me concerns about whether these key modules significantly impact overall performance.

**Essential References Not Discussed:**

The paper overlooks several key studies on attention‐based cropping methods [1,2,3] that are essential for highlighting its contributions.

[1] Chen J, Li H, Liang J, et al. Attention-based cropping and erasing learning with coarse-to-fine refinement for fine-grained visual classification[J]. Neurocomputing, 2022, 501: 359-369.

[2] Wang Y, Zhang Z, Feng L, et al. A new attention-based CNN approach for crop mapping using time series Sentinel-2 images[J]. Computers and electronics in agriculture, 2021, 184: 106090.

[3] Wang W, Shen J. Deep cropping via attention box prediction and aesthetics assessment[C]//Proceedings of the IEEE international conference on computer vision. 2017: 2186-2194.

**Experimental Designs Or Analyses:**

The experimental design is generally sound with thorough ablation studies. However, given that the quality of the attention map significantly impacts performance, evaluating the method with a more advanced VLP model (e.g., BLIP-2) would better validate its robustness.

**Methods And Evaluation Criteria:**

The paper leverages standard zero-shot visual classification and domain generalization benchmarks (i.e., various ImageNet variants, CUB, Oxford Pets, DTD, Food101, and Place365) which effectively assess both global and fine-grained performance.

**Other Comments Or Suggestions:**

My main concern remains the incremental contribution, particularly the limited gains from the attention-based selection compared to the soft matching component. Therefore, I give a "Weak Reject" recommendation and may improve my rating if the rebuttal satisfies.

Update:

I appreciate the authors' efforts in their rebuttal. It effectively addresses most of my concerns. I believe the proposed method is simple but efficient, and as a result, I decide to raise my rating to ``weak accept'' recommendation

**Other Strengths And Weaknesses:**

Strengths

1.	The experiment of this paper is extensive.

2.	It is an interesting idea of attention-based feature selection to enhance the model's performance.

Weaknesses

1.	Limited technical contribution. The proposed ABS closely resembles WCA (ICML, 2024) with only incremental changes, and the Soft Matching module appears to drive most of the performance gains (Please refer to “Relation To Broader Scientific Literature”).

2.	Concerns about the performance on images with multiple objects (e.g., Food101) or complex backgrounds (e.g., Place365). For instance, the Food101 dataset shows lower performance compared to WCA and even negative gains on Place365.

3.	Concerns about the reproducibility. the paper insufficient details on the LLM's prompt generation, alignment process between raw and feature spaces. It is a recommendation for including a pseudo-code.

4.	Formatting issues in some formulas (e.g., Formula 8) that need optimization.

5.	Overall language and clarity could be improved for better readability.

**Questions For Authors:**

Why does ABS perform worse on Food101? Is it related to the complexity of food images, the presence of multiple objects, or another factor?

**Relation To Broader Scientific Literature:**

The paper builds on WCA (ICML, 2024), by incrementally enhancing its pipeline. Whereas WCA uses random cropping and LLM-generated descriptions, this paper integrates attention-based selection to more accurately focus on semantically important regions.

**Theoretical Claims:**

The paper offers only limited theoretical claims without detailed proofs, relying mostly on intuitive arguments.

---

> ### Author Rebuttal · Authors · 2025-03-31
>
> **Q1:** The concerns in Claims And Evidence.
>
> **A1:** Thanks.
> 1. The table below compares applying soft matching alone vs. combined two selections to the baseline. Combined with the ablations in our paper, it shows that individual components yield improvement when used independently, but integration works best. It is noteworthy that the combined two selections outperforms using soft matching alone, proving these key modules significantly enhance overall performance.
> 2. The small improvement observed when using the raw space and feature selection individually because: Using either selection alone may lead to excessive focus on either local or global information. Moreover, employing cropping will focus on specific local regions. While this enables better alignment with LLM descriptions that match the currently focused regions, it weakens the alignment for those unrelated to these regions. Consequently, without soft matching to filter irrelevant descriptions, many unrelated descriptions would adversely affect the current crop's alignment. This phenomenon is also observed in the ablations in WCA, demonstrating that filtering or weighted integration of crops is crucial.
> 3. We replace the components in WCA with ours in the table below. The results prove that our components deliver performance gains compared to the original WCA, proving superiority of our design.
> ||ImageNet|DTD|ImageNetv2|$\Delta$Avg
> |-|-|-|-|-
> |Baseline|69.61|50.53|63.27|
> |Baseline w/ soft m.|70.03|52.89|63.68|+1.06
> |Baseline w/ two sele.|70.32|52.66|64.34|+1.30
> |Ours|**71.92**|**54.26**|**66.19**|+2.98
> |WCA|71.08|52.79|64.71|
> |WCA w/ soft m.|71.37|53.60|65.67|+0.69
> |WCA w/ two sele.|71.42|53.76|65.83|+0.81
>
> **Q2:** More advanced VLP model.
>
> **A2:** Thanks, as your suggestion, we employ the Blip-2 and compare with other baselines. The table below shows that ABS outperform all others across two datasets, demonstrating our effectiveness and adaptability. More advanced VLMs (e.g. ALIGN, AltCLIP and GroupViT) please refer to **our answer A3 to Reviewer dz4u.**
>
> ||Waffle|CuPL|WCA|ABS
> |-|-|-|-|-
> |DTD|45.64|50.74|54.47|**56.12**
> |ImageNet-A|61.43|63.51|72.79|**74.39**
>
> **Q3:** The comment of Broader Scientific Literature.
>
> **A3:** Thanks. Please refer to **our answer A1 to Reviewer bJ6A.**
>
> **Q4:** About attention‐based cropping methods.
>
> **A4:** Thanks. [1] generates attention maps to crop and randomly erasing regions to force the model to focus on key areas. Although their method focus local regions, it fails to address the subsequent loss of global context. In contrast, our approach compensates by using feature selection. We experiment with [1] on our task (as shown in the table below), ABS demonstrate superior performance. This advantage stems from our global semantic compensation for cropped regions.
> ||ImageNet
> |-|-
> |ACEN [1]|61.62
> |Ours|**71.92**
>
> Although [2] and [3] are both attention-related works, [2] uses geographical information, making it only applicable to specific scenarios. [3] requires training for its modules, which contradicts zero-shot tasks. Moreover, neither method considers supplementing global information. In the revision, we will discuss and compare with these methods.
>
> **Q5:** About technical contribution.
>
> **A5:** Thanks. Please refer to **A1 and our answer A1 to Reviewer bJ6A.**
>
> **Q6:** Concern about Food101.
>
> **A6:** Thanks. ABS demonstrates performance improvements across multi-object or complex backgrounds datasets including Place365, only with the exception of Food101.
>
> The Food101 differs from conventional multi-object datasets, it contains inherently multi-label images but provides only single-label annotations. For instance, an image labeled as "french fries" may actually contain multiple objects (e.g., fries, steak, and salad), where non-target objects could occupy a larger visual proportion than the labeled subject (We show several examples in our anonymous link: https://anonymous.4open.science/r/Submission_4487-1B78). Because all these objects fall within the predefined label set, the single-label assignment introduces ambiguity. These inherent properties could cause our method to identify unlabeled object categories within the images.
>
> WCA’s randomness may inadvertently benefit from this properties. However, our experiments confirm substantial improvements in most of classification tasks including multi-object or complex backgrounds datasets.
>
> **Q7:** About prompt and pseudocode.
>
> **A7:** Thanks.
> 1. For the generation of LLM descriptions, we simply follow CuPL, utilizing their publicly released JSON files. The specific prompts can be found in the CuPL.
> 2. For details and pseudocode of the alignment process, please refer to **our answer A2 to Reviewer dz4u and A4 to Reviewer iUpr.**
>
> We will supplement these in future versions of our paper.
>
> **Q8:** Weakness 4&5.
>
> **A8:** Thanks. We will revise our paper thoroughly.
>
> **Q9:** Other comments and questions.
>
> **A9:**  Thanks. Please refer to the answer above.

---

### Decision · Program_Chairs · 2025-05-01

**Decision:**

Accept (poster)

**Comment:**

This paper proposes Attention-Based Selection (ABS), a training-free approach for enhancing zero-shot classification by leveraging attention maps from models like DINO to guide cropping in both raw image and feature space. Combined with a soft matching mechanism to filter LLM-generated descriptions, ABS aims to mitigate the randomness of standard cropping and better integrate local object details with global semantic context. The authors demonstrate state-of-the-art results on several zero-shot and out-of-distribution benchmarks across various VLM backbones. While initial concerns were raised about the incremental nature of the contribution and the effectiveness of individual components, the authors' rebuttal provided convincing additional evidence and clarifications, leading one reviewer to raise their score.

**Strengths:**

* **Novelty & Effectiveness:** The core idea of using attention for dual-space (raw image + feature) cropping combined with soft matching is intuitive and demonstrates strong empirical performance, achieving SOTA results on multiple benchmarks and improving over strong baselines like WCA and CuPL (All Reviewers).
* **Broad Applicability & Robustness:** The method shows effectiveness across various VLM backbones (CLIP, Blip-2, ALIGN, AltCLIP, GroupViT) and exhibits robustness to hyperparameter choices (bJ6A, dz4u, C2Mc).
* **Experimental Rigor:** The paper is supported by extensive experiments, thorough ablation studies validating component contributions, and clear visualizations (C2Mc, bJ6A, iUpr).
* **Clarity & Motivation:** The paper is generally well-written with clear motivation for the proposed approach (bJ6A, dz4u, iUpr).

**Concerns:**

* **Contribution Scope:** Initial concerns about the contribution being incremental over WCA (C2Mc, bJ6A). Authors argued for the novelty of the systematic methodology and dual-space cropping; C2Mc accepted this post-rebuttal, bJ6A acknowledged.
* **Inference Overhead:** The method introduces significant inference time cost due to multiple crops, although authors argue performance gains justify it and cost can be mitigated (iUpr; acknowledged rebuttal).
* **Reliance on Attention Quality:** Performance potentially depends on the quality of the DINO attention maps (iUpr; authors discussed mitigations, reviewer acknowledged).
* **Dataset-Specific Performance:** Initial weaker performance on Food101 was noted (C2Mc; authors provided plausible explanation related to dataset annotation ambiguity, C2Mc accepted).
* **Minor Issues:** Formatting, citation, and clarity issues were noted (All Reviewers; authors promised fixes, reviewers acknowledged).